# Iron Deficiency Is Associated with Elevated Parathormone Levels, Low Vitamin D Status, and Risk of Bone Loss in Omnivores and Plant-Based Diet Consumers

**DOI:** 10.3390/ijms251910290

**Published:** 2024-09-24

**Authors:** M. Pilar Vaquero, Elena García-Maldonado, Angélica Gallego-Narbón, Belén Zapatera, Alexandra Alcorta, Miriam Martínez-Suárez

**Affiliations:** 1Institute of Food Science, Technology and Nutrition (ICTAN), CSIC, 28040 Madrid, Spainangelica.gallego@uam.es (A.G.-N.); bzapatera@ictan.csic.es (B.Z.);; 2Biology Department, Universidad Autónoma de Madrid, 28049 Madrid, Spain

**Keywords:** vitamin D deficiency, bone, bone formation, bone resorption, parathormone, iron status, iron deficiency, vegetarian, vegan, osteoporosis

## Abstract

A cross-sectional study was performed in healthy adults (mean age 28 y, 67% women) whose habitual diet was an omnivore, lacto-ovo vegetarian, or vegan diet. The total sample (*n* = 297) was divided into two groups according to the parathormone (PTH) cut-off value of 65 pg/mL of either normal-PTH (*n* = 228) or high-PTH (*n* = 69). Vitamin D status (25-hydroxycholecalciferol, 25-OHD), PTH, and bone formation (bone alkaline phosphatase, BAP) and bone resorption (N-telopeptides of type I collagen, NTx) markers were determined. Hematocrit, erythrocytes, hemoglobin, platelets, serum iron, serum transferrin, transferrin saturation, and serum ferritin were also measured. In the total sample, 25-OHD and PTH were negatively correlated, and all subjects with high PTH presented vitamin D insufficiency (25-OHD < 75 nmol/L). High bone remodeling was observed in the high-PTH group, with significantly higher NTx and marginally higher BAP compared to the normal-PTH group. Hematocrit and ferritin were significantly lower in the high-PTH compared to the normal-PTH group. However, serum iron was higher in the high-PTH group, which was only observed for the lacto-ovo vegetarian and vegan subjects. It is concluded that both low vitamin D and low iron status are associated with elevated PTH and bone resorption, more in vegetarians than omnivores, which is in line with the hypothesis that chronic iron deficiency in adulthood mainly predisposes to osteoporosis in postmenopausal women and the elderly.

## 1. Introduction

Osteoporosis is an age-related chronic disease that is highly prevalent worldwide but, at the same time, preventable through diet, physical activity, and adequate sunlight exposure. It is a silent disease that develops without apparent symptoms, characterized by weak bones and a high risk of fractures. Its main consequences are fractures along with concomitant pain, disability, and increased comorbidities [1]. The global prevalence of osteoporosis is increasing, which is partly explained by demographic changes and the aging of the population [1,2]. Anemia constitutes another important global health problem, and the main consequences are reduced physical activity, low work performance, increased susceptibility to infections, and impaired thermoregulation and cognitive function [3]. Iron deficiency anemia is the most common leading cause of anemia, principally due to low dietary iron bioavailability, inadequate iron intake during periods of intense growth, and menstrual blood losses in women of fertile age [2].

Bone health is highly related to the process of bone remodeling. Bone remodeling is the coordinated process of bone formation and resorption, resulting from the constant actions of osteoclasts and osteoblasts. Osteoclasts are derived from hematopoietic stem cells and are responsible for resorbing bone and also for attracting bone marrow mesenchymal stem cells that differentiate into osteoblasts, which are the bone-forming cells. Osteoblasts secrete proteins that are needed for renewing the bone organic matrix, such as type I collagen [4,5]. This process is studied by determining biochemical bone markers, which are surrogate markers of osteoblast and osteoclast functions. Among them, bone alkaline phosphatase (BAP) is very sensitive to changes in bone formation, and the cross-linked N-telopeptides of type I collagen (NTx) is a specific marker directly related to bone resorption [6]. The biomarker of vitamin D status is serum 25-hydroxycholecalciferol (25-OHD), and the values represent the sum of the cutaneous synthesis and the ingested amount.

The regulation of bone metabolism is finely tuned by different players. In adult bones, the main regulators are vitamin D and parathormone (PTH). Vitamin D, either produced in the skin from 7-dehydrocholesterol by ultraviolet radiation or provided by the diet, is transported by the D-binding protein to the liver, where it is hydroxylated to produce 25-OHD, which is further hydroxylated in the kidney to yield 1,25-hydrocholecalciferol (1,25-(OH)_2_D). A decrease in circulating calcium due to vitamin D deficiency, inefficient vitamin D metabolism, and/or calcium deficiency triggers the release of PTH, which stimulates renal expression of 1α-hydroxylase to increase the synthesis of 1,25-(OH)_2_D. The main functions of this active form are the enhancement of intestinal calcium absorption, the reduction of urinary excretion of calcium, and the increase in the formation of osteoclasts leading to bone resorption, finally releasing calcium into the blood and restoring calcium levels [7].

There is a metabolic relationship between iron and bone. On the one hand, iron is an essential cofactor for the hydroxylation of prolyl and lysyl residues in procollagen previous to the formation of the triple helix of collagen, which is the main component of the organic bone matrix. It is also required for the activation and inactivation of vitamin D, as these steps are catalyzed by hem-enzymes of the cytochrome P450 superfamily, 25-hydroxylase, 1α-hydroxylase, and 24-hydroxylase [8,9]. On the other hand, vitamin D deficiency can cause reduced erythropoiesis as vitamin D plays a role in cell growth, and several studies have shown that vitamin D supplementation increases erythrocytes, hematocrit, and hemoglobin in humans [10,11].

Evidence of the influence of iron on bone metabolism comes from studies in rats under iron-deficient diets, showing that iron depletion reduces bone mineralization, microarchitecture, and strength by decreasing bone formation [12,13,14,15]. However, there are few studies in humans on the relationship between iron deficiency and bone health. In postmenopausal women, positive associations between dietary iron and bone mineral density (BMD) have been found [16], and in the elderly, BMD was positively associated with hemoglobin levels [17]. One study reported that iron-deficient anemic women presented higher bone resorption compared to controls and that recovery from anemia was accompanied by decreases in bone formation and bone resorption [18]. More recently, vitamin D insufficiency (25-OHD < 75 nmol/L) was associated with over five times increased odds of iron deficiency anemia in undernourished children [19].

The Spanish population is characterized by low vitamin D status compared to Northern countries [20], which is a paradox considering the average number of sunny days per year. This has been partly explained by the low consumption of vitamin D supplements, although other factors, such as avoidance of sun exposure and skin pigmentation, which limits cutaneous synthesis of vitamin D, also play a role.

Our research group has studied in detail the nutritional status of Spanish subjects following omnivorous, lacto-ovo vegetarian, or vegan diets [21,22,23,24]. Vitamin D intake was very low in the three groups and much lower in lacto-ovo vegetarians and vegans than omnivores. It was also observed that vegans ingested less calcium than omnivores and lacto-ovo vegetarians. Consistently, the prevalence of severe vitamin D deficiency was higher in vegans compared to the other two groups, and this group presented higher levels of the bone resorption marker NTx and higher PTH levels compared to omnivores [25]. Concerning iron metabolism, both lacto-ovo vegetarians and vegans presented lower iron status than omnivores. This was attributed to the presence of iron absorption inhibitors in plant-based diets and the additional inhibitory effect of dairy products and egg proteins in the lacto-ovo vegetarian diet [22].

With this background, we hypothesized that low iron status is associated with high bone resorption and risk of bone loss and that this association is stronger in vegetarians than omnivores. Therefore, as part of a larger investigation, the main objective of this cross-sectional study in healthy Spanish adults was to know if iron status differs between those with normal-PTH or high-PTH levels. Additional aims were (1) to assess if there is an association between iron status and PTH and the bone remodeling markers BAP and NTx, (2) if iron status differs between men and women with or without elevated PTH, and (3) if the association between iron and bone biomarkers varies between consumers of plant-based diets and omnivores.

## 2. Results

Table 1 shows the characteristics of the participants. A total of 297 volunteers were recruited and agreed to be enrolled, of which 69 (23%) presented PTH levels above 65 pg/mL, and the remaining had PTH ≤ 65 pg/mL. Women accounted for 67% of the total sample in each of the two PTH groups. Likewise, the distribution of subjects who followed an omnivorous, lacto-ovo vegetarian, or vegan diet was similar in the two PTH status groups, although there was a tendency for a higher proportion of vegans in the high-PTH group (non-significant differences identified by the Pearson Chi-square test). The majority of subjects in the high-PTH group were vitamin D deficient, and the differences between the groups in the prevalence of vitamin D deficiency or insufficiency were highly significant (*p* < 0.001). However, the differences in the prevalence of iron deficiency or iron depletion with or without anemia were not significant.

Other characteristics, such as age, anthropometry, and body composition, did not show differences between the two PTH groups.

Results of PTH, 25-OHD, and the bone remodeling markers according to the PTH status and sex are presented in Table 2. The high PTH group had lower 25-OHD levels (*p* < 0.001) and higher bone resorption marker NTx (*p* = 0.005) but a tendency to higher bone formation marker BAP (*p* = 0.07). Women showed lower BAP levels than men, while the remaining parameters did not show any sex differences.

Concerning the relations between variables, PTH and 25-OHD were inversely related (R^2^ linear = 0.116, *p* < 0.001), with very similar fitted equations for the three diet groups separately. It was observed that except in one case, most of the cases with PTH > 65 pg/mL had 25-OHD levels below sufficiency at 75 nmol/L (Appendix A). In addition, the bone remodeling markers BAP and NTx were positively correlated, and PTH was positively correlated with serum iron (Appendix A).

Table 3 presents the hematological and biochemical variables related to iron metabolism according to PTH status and sex. Hematocrit and ferritin were lower (*p* = 0.039 and *p* = 0.005, respectively), and hemoglobin tended to be lower in the high-PTH group (*p* = 0.058), while serum iron was higher in this group compared with the reference group (*p* = 0.020). Women had lower iron status than men, with significantly lower erythrocytes, hemoglobin, hematocrit, serum iron, transferrin saturation, and ferritin (all, *p* < 0.001) and higher transferrin than men (*p* < 0.001). There were no significant interactions between PTH status and sex.

Appendix A shows additional hematological parameters of the normal-PTH and high-PTH groups, either women or men.

Figure 1 shows the significant differences in iron markers according to PTH status and the diet followed by the participants. It was observed that both factors exerted significant effects. Serum iron was higher in the high-PTH group (*p* = 0.002). However, there was a significant interaction between PTH status and diet; the increase was marked for the lacto-ovo vegetarian and vegan groups but not for the omnivorous group. Concerning serum ferritin, both high PTH (*p* = 0.049) and following a vegetarian diet involved lower ferritin, with lacto-ovo vegetarians and vegans having lower ferritin levels than omnivores (*p* < 0.001).

## 3. Discussion

In this study, the association between bone biomarkers (PTH, vitamin D, bone formation BAP, and bone resorption NTx) and iron status was evaluated in a population of apparently healthy adults who consumed omnivorous or plant-based diets. It was observed that nearly one-fourth of all these individuals presented elevated PTH levels and that almost all of them exhibited vitamin D insufficiency or deficiency. Our results suggest that, according to NTx, bone resorption of the group with high PTH is linked to low vitamin D and iron status.

The inverse association between 25-OHD and PTH was confirmed, and the level of 25-OHD above which the PTH is not elevated is in agreement with previous research in older Spanish subjects [26,27]. Results also show a similar prevalence of hyperparathyroidism in men and women in the three diet groups, although slightly higher in vegans, which can be attributed to the fact that these consumers presented lower intake of essential bone nutrients, such as protein, calcium, and vitamin D [22,25,28].

To explain the high PTH levels, low vitamin D intake should be considered as the main factor. We have previously reported that vitamin D intake was very low in these participants, around 1 µg/day in the two groups consuming plant-based diets and 2 µg/day in the omnivorous group [25]. This amount is insufficient considering the adequate dietary reference intake of 15 µg/day for adults [29]. In fact, as previously reported (25), the mean 25-OHD levels of these volunteers were low even in summer. This finding has been explained by the Mediterranean paradox: although Spain is a sunny country, habits such as avoiding outdoor activities in summer due to elevated temperatures and low use of vitamin D supplements may explain the high rates of vitamin D deficiency.

Another relevant factor to consider is calcium intake, which was below the dietary reference intake [30] in the three diet groups, especially in vegans, due to the lack of consumption of dairy products [25]. Interestingly, elevated intakes of calcium and dairy products have been associated with reduced iron absorption, as an interaction between these foods and iron has been documented. A randomized controlled trial using a skimmed milk product supplemented with iron or iron and vitamin D demonstrated that iron ingested with the product was not bioavailable [10]. Consistently, it has been previously reported that iron status tends to be lower in lacto-ovo vegetarians than in vegans, which was attributed to the inhibiting effect of dairy and also eggs on iron absorption [22]. Therefore, we propose that the low status of vitamin D and iron may be the main determinants of the observed elevated PTH in the studied population.

Participants with high PTH presented low hematocrit and serum ferritin and a tendency to lower hemoglobin and transferrin saturation compared to the reference group. These changes are compatible with lower iron status in the group with high PTH. In this group, the low vitamin D status may contribute to a decrease in the process of erythropoiesis and the formation of red blood cells, which supports present hematological results [10,11]. However, the higher serum iron in this group was unexpected (Table 3). It should be pointed out that the sex influence was studied and was not a confounder, with the proportion of women in the two PTH status groups the same (67%). Moreover, serum iron was similar in omnivores, lacto-ovo vegetarians, and vegans when the PTH classification is not taken into account [22], and in the present study, iron was not elevated in omnivores with high PTH levels (Figure 1).

To explain the results of the iron markers, it should be noted that iron deficiency is generally detected by low erythrocytes, low hemoglobin, low transferrin saturation, low ferritin, and low serum iron. In addition, serum transferrin increases in an attempt to mobilize as much iron as possible to meet the tissue demands. Among all these parameters, serum iron is the least sensitive and is not a good marker of iron status [22,31]. Hemoglobin is the gold standard measurement for the diagnosis of anemia, and cut-off values are established by the World Health Organization (WHO) at <12 g/L for women and <13 g/L for men [3]. Serum ferritin is a marker of iron stores, but it is increased by infection and inflammation, so there is no consensus on cut-off values. The WHO recommends that this marker should only be used to diagnose iron deficiency in apparently healthy individuals [32]. Our research group has consistently used 30 ng/L as the serum ferritin value below which iron deficiency is defined [10,22,24,31]. Therefore, participants whose hemoglobin is within normal ranges but exhibits low ferritin levels are considered to have iron deficiency and a high risk of anemia.

Both vitamin D and iron deficiencies were highly prevalent in these volunteers (Table 1), which was detailed in previous reports [22,24,25]. Figure 2 schematizes the proposed mechanisms for the interaction between iron and bone metabolism. Iron deficiency can induce an increase in bone loss by directly inhibiting the first step of procollagen formation, as the prolyl and lysyl-hydroxylases need Fe(II), leading to a reduction in the formation of collagen fibers and reduced bone formation. Moreover, oxygen is also a cosubstrate in this reaction, and under conditions of iron deficit, hypoxia is common [33]. Hypoxia-inducible factors (HIFs) have a myriad of actions and act in cells derived from bone marrow precursors, such as osteoclasts, finally resulting in bone resorption [34]. However, hypoxia is the main stimulus for erythropoietin (EPO) synthesis, and EPO regulates the formation of bone and stimulates hematopoiesis as well as osteopoiesis in the marrow [35], which could compensate for the inhibitory effect on collagen synthesis. In this line, it was observed that the bone formation marker did not differ between the two PTH status groups, which supports the role of feedback mechanisms.

Another suggested pathway is through the vitamin D activation route (Figure 2). Iron is part of the cytochromes CYP2R1 and CYP27B1 that transform cholecalciferol into the main active metabolite of vitamin D, 1,25-(OH)_2_D. Conversely, in situations of decreased 1,25-(OH)_2_D, PTH is stimulated and increases the renal 1-α hydroxylase to activate the synthesis of this metabolite, resulting in increased bone resorption [7]. Indeed, in the present study, we observed a marked increase in the bone resorption marker in the high-PTH group.

These two metabolic pathways may explain the relationships among iron, vitamin D, and bone, for which there is scientific support in the situations of moderate to severe deficiencies of iron and vitamin D. A comprehensive view of this scheme should include the main regulator of systemic iron, hepcidin, as well as its receptor ferroportin and hemojuvelin that plays a role in hepcidin expression. Hepcidin inhibits iron absorption, decreases the release of iron from macrophages, and releases stored iron from hepatocytes. The hepcidin–ferroportin system is regulated by plasma iron concentrations, hepatic iron stores, erythropoiesis, and inflammation [36]. Usually, either in physiological or pathological iron disorders, serum hepcidin and ferritin levels are positively correlated, while soluble hemojuvelin is inversely related to hepcidin and ferritin. Therefore, under situations of iron deficiency, hepcidin synthesis is minimized, and serum hepcidin levels are negligible. However, hemojuvelin levels might be elevated, although its clinical utility remains to be established [37]. Consequently, ferritin is clinically more useful than hepcidin and hemojuvelin.

It should also be highlighted that bone metabolism is more complex than the simple representation of procollagen synthesis. In the context of the present investigation, the protein ingested by vegetarians may condition an appropriate supply of essential amino acids, such as lysine, which plays an important role in collagen synthesis, and the branched-chain amino acids, which are related to muscular protein maintenance and therefore also contribute to bone strength [25,38].

It is important to note that the participants in this study were not aware of any bone, renal, or hematologic disease. The average age of these adults was nearly 30 years, at which age the skeleton has reached peak bone mass, and bone formation and resorption should be balanced [39,40]. Primary hyperparathyroidism is an endocrine disorder of the parathyroid glands, generally due to a benign overgrowth of parathyroid tissue, in which there is hypercalcemia and inappropriately normal or elevated PTH and is commonly treated by surgery. In contrast, in secondary hyperparathyroidism, the PTH elevation is caused by an external stimulus that causes hypocalcemia [41,42]. The results of this study are consistent with asymptomatic secondary hyperparathyroidism. Concerning iron pathologies, iron deficiency anemia was negligible, but iron stores were low, indicating moderate iron deficiency. This was more pronounced in women, mainly because of the known influence of menstrual blood loss on iron status [43,44].

The present results suggest that bone remodeling is accelerated when there is vitamin D and iron insufficiency and that chronic deficiency of both nutrients can lead to bone loss. This study adds new data in line with the hypothesis that chronic iron deficiency during adulthood predisposes to osteoporosis [4,9]. However, this hypothesis should be confirmed by future studies. The modulating effect of the diet (omnivorous, lacto-ovo vegetarian, or vegan) appears to be weak, but plant-based diet consumers whose PTH levels were above normal had elevated serum iron, although without reaching the omnivore threshold, which warrants further investigation.

Future studies are needed. The influence of iron deficiency without other nutritional deficiencies, such as vitamin D deficiency and inadequate calcium and protein intake, on bone deserves new research. In addition, it is important to investigate the molecular mechanisms behind the effects. It is not known whether the influence of moderate iron deficiency or severe anemia on bone is similar or not because hypoxia induces profound metabolic responses. In addition, the genetic background and the involvement of the iron-bone relationship in many pathological conditions should be considered.

This study has several limitations. The design was cross-sectional, and no cause-and-effect relationship can be established. Serum ionic calcium was not measured, which could have been useful in detecting cases of primary hyperparathyroidism. Hepcidin levels were also not measured, although, under conditions of iron deficiency, they were expected to be very low. Renal function was also not measured. However, according to the inclusion criteria for this study, the participants were apparently healthy, younger than 45 years of age, and once selected, none of the subjects reported any renal disease. The strengths of the study are the large sample size, the standardization of all measurements performed with the same techniques, the inclusion of both sexes and the study of diet type.

## 4. Materials and Methods

### 4.1. Study Design

The design is a cross-sectional study, part of a larger project on the health status of adults consuming omnivorous or vegetarian diets. Recruitment and laboratory analysis were performed in the Madrid region of Spain (latitude 40°24′59.4″ N) during the summer of 2017, winter of 2020, and summer of 2021. Inclusion criteria were healthy adults (age ≥ 18 years) consuming omnivorous, lacto-ovo vegetarian, or vegan diets for at least 6 months. Exclusion criteria were minors, including following his/her current diet for less than 6 months, eating disorders, diagnosis of digestive, hematological, endocrine, renal, or oncological diseases, pregnancy, breastfeeding, menopause, having donated blood in the 3 months prior to the study, and for the lacto-ovo vegetarian and vegan participants occasional consumption of meat or fish.

Each participant signed an informed consent form before the start of the study. The study was approved by the Ethics Committees of CSIC (Cod. 104/2019, date 19 December 2019 and modification of Cod. 46/2021, date 5 March 2021) and the Hospital Puerta de Hierro, Majadahonda, Spain (Cod. PI 176/19, date 18 November 2019, and modification of Cod. PI 176/19, date 2 March 2021).

### 4.2. Determinations

Body weight, height, and waist and hip perimeters were measured by a trained member of the research group using standardized procedures, and body mass index (BMI) was calculated. Bone mass, muscle mass, and fat mass were determined by a body composition monitor (Tanita BC-601, Tanita Ltd., Amsterdam, The Netherlands). The precision of these measurements was <0.1%.

Participants attended the Human Nutrition Unit of the institute between 7:45 h and 9:30 h after a 10–12 h fasting period. Blood samples were collected by venipuncture, and the fasting second-morning urine was also collected just before or after blood sampling. Total blood was used for hematological analyses, and serum was separated by centrifugation (centrifuge Jouan CR-312, Ilkeston, UK) for 15 min at 1000× *g*. Aliquots of serum and urine were stored at −80 °C for further analyses.

Hematocrit, erythrocytes, hemoglobin, mean corpuscular volume (MCV), red cell distribution width (RDW), mean corpuscular hemoglobin (MCH), mean corpuscular hemoglobin concentration (MCHC), platelets, mean platelet volume, serum iron, serum transferrin, transferrin saturation, and serum ferritin were measured by automatic analyzers.

Parathyroid hormone (PTH) was analyzed in serum by ELISA using the kit PTH Parathyroid Intact EIA from DRG (DRG Instruments GmbH, Marburg, Germany). Serum 25-hydroxyvitamin D (25-OHD) and the bone formation marker (bone alkaline phosphatase, BAP) were determined by ELISA kits 25-hydroxyvitamin D EIA and Ostase BAP EIA, commercialized by Immunodiagnostic Systems (IDS, Boldon, UK). The bone resorption marker, NTx, was determined using the ELISA kit NTx Urine Osteomark (Alere Scarbourgh Inc., Scarborough, ME, USA). Urine creatinine was measured using an autoanalyzer (Olympus AU5800 Beckman, Beckman, Nyon, Switzerland).

All bone turnover markers were analyzed using one single batch for each marker. The intra- and inter-assay coefficients of variation were as follows: PTH, 2.7 and 4.1%; 25-OHD, 2.3% and 8.2%; BAP, 2.8% and 6.4%; and NTX, 4.1% and 3.4%. NTx is expressed as nanomoles of bone collagen equivalents per millimole of creatinine (nM BCE/mM Cr). All determinations comply with the standard ISO 9001:2015 requirements [45].

### 4.3. Cut-Off Values

The following cut-off values were used:PTH status: PTH > 10 and ≤65 pg/mL, normal; PTH > 65 pg/mL, high.Vitamin D status: 25-OHD ≥ 75 nmol/L, sufficiency; 25-OHD < 75 nmol/L and ≥50 nmol/L, insufficiency; 25-OHD < 50 nmol/L, deficiency.Iron status: ferritin > 30 ng/mL, sufficiency; ferritin ≤ 30 and >15 ng/mL, deficiency; ferritin ≤ 15 ng/mL, depletion; hemoglobin < 12 g/dL for women and <13 g/dL for men, or use of iron supplements, iron deficiency anemia.

### 4.4. Statistical Analyses

Results are expressed as mean ± SD, except those of categories that are presented as numbers (%). Variable distributions were checked, and ferritin, BAP, and NTx were log-transformed before statistical analyses.

Subjects were classified according to PTH status using the cut-off value of 65 pg/mL (normal-PTH, high-PTH), sex (women, men), and diet (omnivore, lacto-ovo vegetarian, vegan). General linear models were applied to test the effect of the factors PTH status, sex, diet, and the interactions between PTH status and sex and PTH status and diet on the dependent variables (GLM multivariate procedure). Categorical data were analyzed by Pearson Chi-Square tests. The Bonferroni correction for multiple comparisons was used.

The data were analyzed using SPSS for Windows version 29.0 (IBM SPSS Statistics for Windows, Armonk, NY, USA). The level of significance was set at *p* < 0.05.

## Figures and Tables

**Figure 1 ijms-25-10290-f001:**
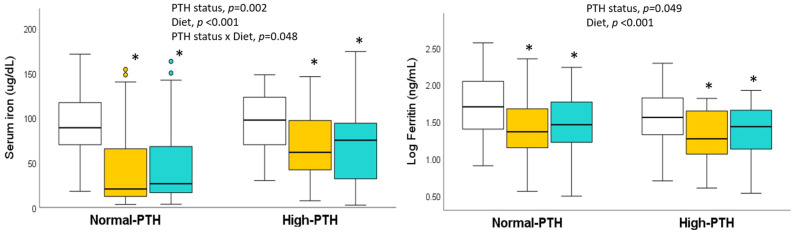
Serum iron and log-ferritin, according to PTH status and diet. Boxplots show the median with the 25th and 75th percentiles and the minimum and maximum values. Boxes with different colors indicate 
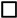
 omnivores, 
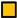
 lacto-ovo vegetarians, and 
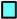
 vegans. * Significantly different compared to omnivores (general linear model with Bonferroni correction for multiple comparisons).

**Figure 2 ijms-25-10290-f002:**
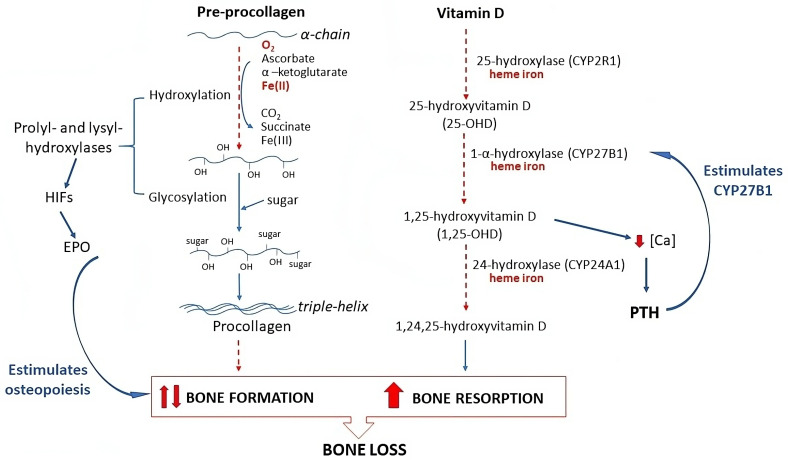
Proposed mechanisms of the relationship between both iron and vitamin D deficiencies and risk of bone loss in the present study. HIFs, hypoxia-inducible factors; EPO, erythropoietin; red lines, inhibition; blue lines, activation; blue curve lines, feedback regulation. Iron deficit (left cascade) at the rough endoplasmic reticulum can reduce the enzymatic activity of prolyl- and lysyl-hydroxylases that catalyze the first step of procollagen synthesis from the α-chains of pre-collagen. Shortage of oxygen in the same reaction stimulates HIFs, which can induce osteoclast formation in the bone marrow, but it is possible that the main effect is to secrete EPO by the renal interstitium, which is a powerful hormone that activates the hematopoiesis and osteopoiesis in the bone marrow. Under conditions of vitamin D insufficiency (right cascade) due to low intake or low sunlight exposure, additional iron deficiency reduces the formation of the main active metabolite of vitamin D, 1,25-(OH)2D, by reducing the 25- and 1α-hydroxylations, and also the final 24-hydroxylation that involves catabolism of the vitamin D. A decrease in 1,25-(OH)2D limits intestinal calcium absorption and lowers ionic calcium in the circulation, which is the main stimulus for PTH secretion. PTH activates the renal expression of the 1α-hydroxylase, which increases 1,25(OH)2-D levels to restore ionic calcium by increasing intestinal calcium absorption, reducing its urinary excretion, and increasing bone resorption.

**Table 1 ijms-25-10290-t001:** Age, diet, vitamin D status, iron status, and anthropometric and body composition characteristics of participants, according to PTH status.

	Normal-PTH (≤65 pg/mL)	High-PTH (>65 pg/mL)	*p*
*n* (%)	228 (76.8)	69 (23.2)	
Women, *n* (%)	153 (67.1)	46 (66.7)	0.946
Omnivore, *n* (%)	71 (31.1)	18 (26.1)	
Lacto-ovo vegetarian, *n* (%)	77 (33.8)	18 (26.1)	0.158
Vegan, *n* (%)	80 (35.1)	33 (47.8)	
Vitamin D sufficiency ^a^, *n* (%)	19 (8.3)	1 (1.5)	
Vitamin D insufficient ^b^, *n* (%)	88 (38.6)	12 (17.6)	**<0.001**
Vitamin D deficiency ^c^, *n* (%)	121 (53.1)	55 (80.9)	
Iron sufficiency ^d^, *n* (%)	112 (49.1)	29 (42.0)	
Iron deficiency ^e^, *n* (%)	69 (30.3)	20 (29.0)	0.327
Iron depletion and anemia ^f^, *n* (%)	47 (20.6)	20 (28.9)	
Age (y)	28 ± 7	27 ± 6	0.270
Body weight (kg)	62.6 ± 11.5	62.6 ± 10.4	0.870
BMI (kg/m^2^)	22.4 ± 3.3	22.2 ± 2.7	0.738
Waist perimeter (cm)	78.6 ± 9.2	76.7 ± 8.8	0.234
Hip perimeter (cm)	95.4 ± 6.8	94.9 ± 5.5	0.609
Muscle mass (kg)	46.7 ± 9.4	46.7 ± 9.3	0.858
Fat mass (kg)	21.9 ± 7.8	21.5 ± 7.5	0.625
Bone mass (kg)	2.5 ± 0.5	2.5 ± 0.4	0.970

Values are *n* with % or mean ± SD. PTH, parathormone; BMI, body mass index; ^a^ 25-OHD ≥ 75 nmol/L; ^b^ 25-OHD < 75 and ≥50 nmol/L; ^c^ 25-OHD < 50 nmol/L. ^d^ ferritin > 30 ng/mL; ^e^ ferritin 16–30 ng/mL; ^f^ ferritin ≤ 15 ng/mL or anemia. Differences between groups in the proportions of subjects were analyzed by the Pearson Chi-square test. Differences in the continuous variables were analyzed by the T test. *p* in bold indicates significant effects.

**Table 2 ijms-25-10290-t002:** Parathormone, vitamin D status, and bone remodeling of participants according to PTH status and sex.

	Normal-PTH (≤65 pg/mL)			High-PTH (>65 pg/mL)				
	Man *n* = 75	Woman *n* = 153	All *n* = 228	Man *n* = 23	Woman *n* = 46	All *n* = 69	*p* Sex	*p* PTH Status
PTH (pg/mL)	42.0 ± 13.5	42.2 ± 13.8	42.1 ± 13.7	79.4 ± 12.7	80.4 ± 11.3	80.1 ± 11.7	0.754	**<0.001**
25-OHD (nmol/L)	47.4 ± 16.8	50.2 ± 19.3	49.3 ± 18.5	37.7 ± 15.7	38.5 ± 13.4	38.2 ± 14.1	0.395	**<0.001**
BAP (µg/L) ^a^	18 (17, 20)	14 (14, 15)	15 (14, 17)	21 (17, 27)	16 (15, 18)	17 (15, 19)	**0.004**	0.070
NTx (nmol/mmol creatinine) ^a^	76 (69, 86)	69 (62, 69)	68 (66, 74)	100 (90, 114)	93 (76, 109)	94 (82, 108)	0.468	**0.005**

Values are mean ± SD, except ^a^ that is median and 95% CI; *p* in bold indicates significant effects. PTH, parathormone; 25-OHD, 25-hydroxycholecalciferol; BAP, bone alkaline phosphatase; NTx, N-terminal telopeptide of collagen I. There were no significant sex–PTH status interactions (general linear model with the Bonferroni correction for multiple comparisons).

**Table 3 ijms-25-10290-t003:** Hematological and biochemical parameters of participants according to sex and PTH status.

	Normal-PTH (≤65 pg/mL)			High-PTH (>65 pg/mL)				
	Man *n* = 75	Woman *n* = 153	All *n* = 228	Man *n* = 23	Woman *n* = 46	All *n* = 69	*p* Sex	*p* PTH Status
Erythrocytes (10^6^/mm^3^)	5.1 ± 0.4	4.5 ± 0.3	4.7 ± 0.4	5.1 ± 0.4	4.4 ± 0.3	4.6 ± 0.5	**<0.001**	0.340
Hemoglobin (g/dL)	15.5 ± 1.0	13.5 ± 1.0	14.2 ± 1.4	15.2 ± 1.1	13.2 ± 1.1	13.9 ± 1.4	**<0.001**	0.058
Hematocrit (%)	45.5 ± 2.8	40.6 ± 2.9	42.2 ± 3.7	44.9 ± 3.2	39.5 ± 3.1	41.3 ± 4.0	**<0.001**	**0.039**
Serum iron (µg/dL)	79.6 ± 48.5	51.0 ± 44.1	60.4 ± 47.5	93.5 ± 39.8	67.4 ± 39.4	76.1 ± 41.2	**<0.001**	**0.020**
Transferrin (mg/dL)	260 ± 41	300 ± 57	287 ± 56	279 ± 48	296 ± 58	290 ± 55	**<0.001**	0.323
Transferrin saturation (%)	32.3 ± 11.8	23.1 ± 13.5	26.1 ± 13.6	28.3 ± 11.1	20.6 ± 11.3	23.2 ± 11.8	**<0.001**	0.076
Ferritin (ng/mL) ^a^	83 (64, 108)	23 (20, 26)	30 (26, 33)	51 (36, 65)	19 (15, 27)	27 (20, 36)	**<0.001**	**0.005**

Values are mean ± SD, except ^a^ that is median and 95% CI; *p* in bold indicates significant effects. There were no significant sex–PTH status interactions (general linear model with Bonferroni correction for multiple comparisons).

## Data Availability

The data can be shared by request directed to the corresponding author.

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
