# Peer review of "Iron Deficiency Is Associated with Elevated Parathormone Levels, Low Vitamin D Status, and Risk of Bone Loss in Omnivores and Plant-Based Diet Consumers"

_ijms, 2024, doi:10.3390/ijms251910290_

Round 1

Reviewer 1 Report

Comments and Suggestions for Authors

In the publication submitted to IJMS in the section Biochemistry of the Special issue: Vitamins and Derived Cofactors: Transport, Metabolism, Assembly and Deficiencies, entitled: Low iron and vitamin D status are associated with high parathormone levels and increased risk of bone loss in healthy omnivores and vegetarians by: M Pilar Vaquero, Elena García-Maldonado, Angélica Gallego-Narbón, Belén Zapatera, Alexandra Alcorta, Miriam Martínez-Suárez authors present a cross-sectional study conducted on healthy volunteers in Spain following different diets, i.e. omnivores, vegetarians and ovo-vegetarians. The aim of this study is to attempt to link iron metabolism with bone metabolism. Then, iron metabolism was determined in women and men depending on PTH concentration. Finally, the effect of the diet on the levels of iron, vitamin D and PTH was assessed. The reader has the impression that the goals of this work are not identical with the research hypothesis, but were created ad hoc while writing the manuscript. The aims of the work are so extensive that it is not clear what the purpose of the work is and what the research assumptions and hypotheses are.

At the beginning of the review, I would like to emphasize that the scope of the research does not predispose the manuscript to publication in IJMS due to the too descriptive and completely non-molecular nature of the research. Similar studies were published in: Ambroszkiewicz J, Gajewska J, Mazur J, Kuśmierska K, Klemarczyk W, Rowicka G, Strucińska M, Chełchowska M. Dietary Intake and Circulating Amino Acid Concentrations in Relation with Bone Metabolism Markers in Children After Vegetarian and Omnivorous Diets. Nutrients. 2023 March 12; 15 (6): 1376. doi: 10.3390/nu15061376. PMID: 36986105; PMCID: PMC10055473.

however, they represent a much better approach to the topic, and the journal Nutrients also focuses on this type of research. It is a great pity that this work was not cited in the manuscript. The authors of the work presented to me for review also published studies very similar to those submitted for review in IJMS in Nutrients, (García-Maldonado E, Gallego-Narbón A, Zapatera B, Alcorta A, Martínez-Suárez M, Vaquero MP. Bone Remodelling, Vitamin D Status, and Lifestyle Factors in Spanish Vegans, Lacto-Ovo Vegetarians, and Omnivores. 16(3):448. Here, too, the work by Ambroszkiewicz et al. was not cited. (2023), and it's worth doing. Moreover, I must say that the regulation of iron metabolism in relation to bone metabolism is a quite well-studied topic, especially in the case of CKD, thalassemia, anemia and hemochromatosis. When considering iron metabolism alone, one cannot rely solely on serum ferritin (derived from liver stores in physiological states) or serum iron. Importantly, the ranges in which level changes occur are within normal limits and at the beginning of the second quartile. Moreover, in many cases, TIBC and TFsat indicate the limits of anemia (especially in women), even though the Hb level has fallen below the anemic level.

I have additional comments regarding the research conducted, the analysis and discussion of which is necessary in the text.

1. The title of the work is misleading because regardless of the PTH level, the iron and ferritin levels in vegetarians are always lower than in omnivores. It is nothing new and nothing groundbreaking that problems with iron metabolism resulting from its supply are common

2. The research hypothesis and the authors' goals should be provided. What do they want to convey to the reader? The conclusions are speculative and do not result from research results.

3. The most important thing that prevents drawing conclusions are shortcomings/limitations resulting from the methodology (calcium, serum phosphorus, dietary interview, gender balance, young and very healthy volunteers), which the authors warn the reviewer about. But that doesn't change anything.

4. Finally, Figure 3 actually shows what could be proven if the authors went deeper.

Comments on the Quality of English Language

n/a

Author Response

In the publication submitted to IJMS in the section Biochemistry of the Special issue: Vitamins and Derived Cofactors: Transport, Metabolism, Assembly and Deficiencies, entitled: Low iron and vitamin D status are associated with high parathormone levels and increased risk of bone loss in healthy omnivores and vegetarians by: M Pilar Vaquero, Elena García-Maldonado, Angélica Gallego-Narbón, Belén Zapatera, Alexandra Alcorta, Miriam Martínez-Suárez authors present a cross-sectional study conducted on healthy volunteers in Spain following different diets, i.e. omnivores, vegetarians and ovo-vegetarians. The aim of this study is to attempt to link iron metabolism with bone metabolism. Then, iron metabolism was determined in women and men depending on PTH concentration. Finally, the effect of the diet on the levels of iron, vitamin D and PTH was assessed. The reader has the impression that the goals of this work are not identical with the research hypothesis, but were created ad hoc while writing the manuscript. The aims of the work are so extensive that it is not clear what the purpose of the work is and what the research assumptions and hypotheses are.

Thank you for four comments that are very useful for the improvement of our manuscript.

We agree with the reviewer that the hypothesis and objectives of our manuscript had not been clearly defined and thus we have improved them in this revised version (lines 100-106).

At the beginning of the review, I would like to emphasize that the scope of the research does not predispose the manuscript to publication in IJMS due to the too descriptive and completely non-molecular nature of the research. Similar studies were published in: Ambroszkiewicz J, Gajewska J, Mazur J, Kuśmierska K, Klemarczyk W, Rowicka G, Strucińska M, Chełchowska M. Dietary Intake and Circulating Amino Acid Concentrations in Relation with Bone Metabolism Markers in Children After Vegetarian and Omnivorous Diets. Nutrients. 2023 March 12; 15 (6): 1376. doi: 10.3390/nu15061376. PMID: 36986105; PMCID: PMC10055473.

however, they represent a much better approach to the topic, and the journal Nutrients also focuses on this type of research. It is a great pity that this work was not cited in the manuscript. The authors of the work presented to me for review also published studies very similar to those submitted for review in IJMS in Nutrients, (García-Maldonado E, Gallego-Narbón A, Zapatera B, Alcorta A, Martínez-Suárez M, Vaquero MP. Bone Remodelling, Vitamin D Status, and Lifestyle Factors in Spanish Vegans, Lacto-Ovo Vegetarians, and Omnivores. 16(3):448. Here, too, the work by Ambroszkiewicz et al. was not cited. (2023), and it's worth doing.

We understand your doubt regarding meeting the scope of the journal. However, we were invited by the Editors who consider that our research is appropriate for the especial issue entitled “Vitamins and Derived Cofactors: Transport, Metabolism, Assembly and Deficiencies”.

We have incorporated the work of Ambroszkiewicz et al. (2023) that supports the role of essential amino acids to the discussion (paragraph lines 287-292) and bibliography (Ref. 37, revised version), and agree with the reviewer of its great interest.

Moreover, I must say that the regulation of iron metabolism in relation to bone metabolism is a quite well-studied topic, especially in the case of CKD, thalassemia, anemia and hemochromatosis. When considering iron metabolism alone, one cannot rely solely on serum ferritin (derived from liver stores in physiological states) or serum iron. Importantly, the ranges in which level changes occur are within normal limits and at the beginning of the second quartile. Moreover, in many cases, TIBC and TFsat indicate the limits of anemia (especially in women), even though the Hb level has fallen below the anemic level.

We are grateful to your comments. However, we tried to focus the discussion on iron deficiency of apparently healthy adults. Thus, the link between iron overload, e.g. hemochromatosis, and bone is not explained, neither the possible mechanisms in other pathologies such as CKD.

A new paragraph has been added at the end of the discussion (lines 276-286), and another paragraph has been added mentioning that future studies are needed (lines 313-319), taken also into account comments by referee 2.

I have additional comments regarding the research conducted, the analysis and discussion of which is necessary in the text.

  1. The title of the work is misleading because regardless of the PTH level, the iron and ferritin levels in vegetarians are always lower than in omnivores. It is nothing new and nothing groundbreaking that problems with iron metabolism resulting from its supply are common.

Thank you. We have modified the title. Please note that the results show that omnivores are not clearly separated from the other two groups regarding the association of iron markers with PTH and bone remodeling markers.

  1. The research hypothesis and the authors' goals should be provided. What do they want to convey to the reader? The conclusions are speculative and do not result from research results.

The hypothesis has been added before the objectives, and the third objective has been reworded (lines 100-105). This is a descriptive study and no cause-and-effect conclusion can be obtained from it (line 320). To clarify this, the conclusion paragraph has been slightly modified (lines 305-312).

  1. The most important thing that prevents drawing conclusions are shortcomings/limitations resulting from the methodology (calcium, serum phosphorus, dietary interview, gender balance, young and very healthy volunteers), which the authors warn the reviewer about. But that doesn't change anything.

The conclusion of the abstract has been slightly modified (lines 23-24).

The conclusions of the discussion have been modified as follows (lines 303-312):

Present results suggest that bone remodeling is accelerated when there is vitamin D and iron insufficiency and that chronic deficiency of both nutrients can lead to bone loss. This study adds new data in line of the hypothesis that chronic iron deficiency during adulthood predisposes to osteoporosis [4,9]. However, this hypothesis should be confirmed by future studies. The modulating effect of the usual diet (omnivorous, lacto-ovo vegetarian or vegan) appears to be weak, but plant-based diet consumers whose PTH levels were above normal had elevated serum iron, although without reaching the omnivore threshold, which warrants further investigation.

The submitted version was:

Present results confirm that bone remodeling is accelerated when there is vitamin D and iron insufficiency and suggest that chronic deficiency of both nutrients can lead to bone loss. This study supports the hypothesis that chronic iron deficiency during adulthood predisposes to osteoporosis (4,9).

Therefore, our results suggest that the combination of mild iron deficiency with vitamin D insufficiency may lead to PTH elevation and risk of bone loss. The modulating effect of the usual diet (omnivorous, lacto-ovo vegetarian or vegan) appears to be weak, but plant-based diet consumers whose PTH levels were above normal had elevated serum iron, although without reaching the omnivore threshold, which warrants further investigation.

  1. Finally, Figure 3 actually shows what could be proven if the authors went deeper.

Yes, we agree that this is the most important figure and should be seen as ideas for future in-depth investigations. Thank you.

Reviewer 2 Report

Comments and Suggestions for Authors

The presented manuscript is a valuable source of new information on the pathogenesis of osteoporosis, its connection with iron, vitamin D, and PTH. Nevertheless, I am attaching my comments to the manuscript:

1. I have no comments on the abstract and introduction.

2. The aim of the study has been clearly defined.

3. Please add a column with p-values to Table 1 and appropriately comment on these results.

4. The notation "ferritin ≤ 30 and >15 ng/mL" - wouldn't it be better to write "ferritin 15-30 ng/mL"?

5. The notation "that are median and 95% CL" should be "that are median and 95% CI."

6. The figures are of poor quality.

7. In Figure 2, there should be a legend with colors; the description alone is not enough.

8. Discussion - please provide future perspectives - how other parameters of iron metabolism might be significant in the described processes. Please discuss hepcidin (doi: 10.3390/ijms22126493) and sHJV (doi: 10.3390/cancers15041041). Please use the provided literature.

Author Response

Thank you very much for your positive opinion on our manuscript and your valuable comments. Bellow, you we respond to your points

  1. I have no comments on the abstract and introduction.

Thank you

  1. The aim of the study has been clearly defined.

Thank you

  1. Please add a column with p-values to Table 1 and appropriately comment on these results.

Added. Please note that the % values were analyzed by Pearson Chi-Square tests.

  1. The notation "ferritin ≤ 30 and >15 ng/mL" - wouldn't it be better to write "ferritin 15-30 ng/mL"?

We agree that it is easier the expression “ferritin 15-30”, however it is more correct "ferritin ≤ 30 and >15 ng/mL" , because the value ferritin=15 is not included in this class but ferritin=30 is included. We have decided to use the simplified expression in the table 1 foot and to maintain the notation as it was in the original manuscript in the Methods section (lines 383-384).

  1. The notation "that are median and 95% CL" should be "that are median and 95% CI."

Thank you. It was a mistake.

  1. The figures are of poor quality.

The quality of the figures have been improved and in addition there are given as separated mages for production.

  1. In Figure 2, there should be a legend with colors; the description alone is not enough.

We have added the colors in the legend.

  1. Discussion - please provide future perspectives - how other parameters of iron metabolism might be significant in the described processes. Please discuss hepcidin (doi: 10.3390/ijms22126493) and sHJV (doi: 10.3390/cancers15041041). Please use the provided literature.

We have added a new paragraph at the end of the discussion and the corresponding literature including the review that you provided us with (lines 276-286).

We have modified the paragraph of the conclusions at the end of the discussion and mentioned that more research is needed on the molecular bases involved (lines 313-316).

Finally, we added in the limitations of the study a sentence indicating that hepcidin was not measured (lines 322-333). Thank you.

Reviewer 3 Report

Comments and Suggestions for Authors

The authors of the publication entitled ‘Iron deficiency is associated with elevated parathormone levels and increased risk of bone loss in omnivores and vegetarians with low vitamin D status present a very interesting cross-sectional study.

General comments

In general, this publication investigated quit a large number of blood samples including various biomarkers of healthy female and male adults in connection to their habitual diet at one time point.

(1.)  However, the title suggests (…’increased risk of bone loss…’) that the markers have been sampled more than at one time point. Furthermore, the primary aim of this project is not clearly stated. If considering the PICO method, what was the primary aim and what were secondary objectives?

(2.)  I’m not able to follow your presentation of the results… Why did you not correlate your primary endpoint parameter (e.g., PTH as actual value) to all other measured samples and include them as e.g., supplement table and systematically present the most important findings in the result section (all other analyses could be presented as supplement material)?

The publications would much improve by a more clear and systematically elaboration of the results.

Detailed comments

Please find all comments and suggestions in the pdf file!

Comments on the Quality of English Language

Please find all comments and suggestions in the pdf file!

Author Response

General comments

Thank you for your helpful comments and your opinion on our study.

(1) We have changed the title to avoid misunderstanding because this was a cross-sectional study and the measurements were taken at only one time point. PTH was our main variable and the primary aim was to compare the iron biomarkers of the two groups with normal PTH and high PTH. In the revised manuscript, the population, study design, primary and secondary objectives are more clearly presented at the end of the Introduction.

(2) The correlations are given in Supplementary materials as Figure 1S and Table 1S. Accordingly, the Results section has been modified and shortened.

Detailed comments

Thank you for providing your suggestions in the pdf file, this was really useful. You will find our responses below each of your comments in the attached file

Round 2

Reviewer 1 Report

Comments and Suggestions for Authors

Dear authors, I regret to reject your manuscript. The presented research results are still incomplete, which means that the manuscript has many shortcomings and, most importantly, requires supplementing with new data. I hope other reviewers also recognize my reservations. Kind regards.P

Author Response

Thank you for your time and effort in reviewing our manuscript. We respect your opinion, but we are very surprised by your final decision, because we think that we have answered all your questions adequately and honestly, and we think that the aims and methods are good, and therefore the obtained results have scientific quality.

Reviewer 2 Report

Comments and Suggestions for Authors

The authors have significantly improved the manuscript and have positively addressed most of the reviewers' comments. The section in the discussion regarding hepcidin must relate to the results, and in fact, the authors should suggest a future research direction based on hepcidin, but also on hemojuvelin. Therefore, they should also modify the literature to include the references I originally proposed. Aside from that, I have no critical remarks; I still consider the manuscript to be carefully prepared and valuable from a clinical point of view.

Author Response

Please accept our apologies for the omission of the hemojuvelin reference. We have rectified this in the revised version (lines 278-288, ref num. 37).

We are grateful for your insights on the clinical value of our study.

We appreciate your time and feedback on our manuscript.

Round 3

Reviewer 1 Report

Comments and Suggestions for Authors

Dear Authors. Of course you have the right to disagree with my opinion. Let me stick to my opinion in this case. I think that additional analyzes and designing the research in such a way that it is possible to compare research groups according to the dose of iron used will significantly increase the scientific value of the article. The other reviews you received probably confirmed your opinion. I stand by my decision, which is negative. Kind regards.

Author Response

Thank you for your time spent in reviewing our manuscript. We will follow Editors recommendations.

Reviewer 2 Report

Comments and Suggestions for Authors

The authors have addressed all my questions and concerns.

Author Response

Thank you very much.